# Torsion is a dynamic regulator of DNA replication stalling and reactivation

Xiaomeng Jia [1,2], Xiang Gao[1,2], Shuming Zhang[1,2], James T. Inman[1,2], Yifeng Hong [3], Anupam Singh [4], Fahad Rashid [5], James M. Berger [5], Smita S. Patel[4] & Michelle D. Wang [1,2] ✉

DNA's helical structure necessitates replisome rotation relative to DNA during replication, creating inevitable topological challenges. How replication generates and overcomes torsional stress remains unclear. Here, we developed a high-resolution, label-free, real-time assay to track DNA rotation by T7 replisome and its slowing under torsional stress. While helicase or DNA polymerase (DNAP) alone is a weak rotary motor, together they form the most powerful DNA rotary motor yet studied, generating ~22 pN·nm torque before stalling, twice that of *E. coli* RNA polymerase. Upon stalling, helicase-DNAP interactions stabilize the fork; without them, regression can extend hundreds of base pairs. Prolonged stalling inactivates the replisome, but excess DNAP, aided by interactions with helicase, promotes restart. Gyrase supports steady replication and enables timely restart of stalled forks. These findings demonstrate that helicase-DNAP synergy is essential for maintaining fork integrity under torsion, and that torsion is a key regulator of replication stalling and reactivation.

Upon the discovery that DNA is a right-handed double helix, Watson and Crick immediately called attention to the formidable topological challenges encountered by DNA replication[1,2]. The helical nature of DNA dictates that rotational motion is inherent to replication–for every 10.5 bp (helical pitch) of DNA replicated, the replisome must rotate one turn relative to the parental DNA, leading to over-twisting of the DNA (Fig. 1a). The resulting (+) DNA supercoiling cannot dissipate fully at the distal ends of the DNA[3–5] and must be relaxed by topoisomerases, essential enzymes for topological resolution[6]. However, even the native complement of topoisomerases is insufficient to fully relieve torsional stress across the genome[7–17]. During replication, torsional stress is found during elongation[18–22], near termination[23–26], at DNA fragile sites[27], and at conflicts with other motor proteins, such as an RNA polymerase[28–31]. This indicates that topoisomerases cannot always keep up with the torsional load of genome replication[32,33].

Replication generates torsion, which, in turn, may regulate replication. Unlike stress from local obstacles (such as bound proteins and DNA lesions), torsion acts over distance and can impact regions separated by thousands of base pairs[29]. Torsion ahead of a replisome may dissociate bound proteins[34] or stall oncoming motors[35–37]; torsion behind a replisome can entangle daughter DNA strands, hindering chromosome segregation[18–27,38–40]. Importantly, excessive torsion can lead to replication fork stalling, but how individual motors in the replisome impact this process and the role they play in subsequent fork restart and/or DNA damage repair remains enigmatic.

Because of DNA's inherent helical nature, replication under torsion is a fundamental problem in biology. However, this problem is exceedingly complex, presenting significant challenges for conceptualization and experimentation. There has been little mechanistic understanding of how a replisome elongates against torsion and how this torsion, in turn, impacts replication; this is mostly due to a lack of

[1]Howard Hughes Medical Institute, Cornell University, Ithaca, NY, USA. [2]Department of Physics & LASSP, Cornell University, Ithaca, NY, USA. [3]Department of Electrical and Computer Engineering, Cornell University, Ithaca, NY, USA. [4]Department of Biochemistry and Molecular Biology, Robert Wood Johnson Medical School, Rutgers University, Piscataway, NJ, USA. [5]Department of Biophysics and Biophysical Chemistry, Johns Hopkins University School of Medicine, Baltimore, MD, USA. ✉e-mail: mwang@physics.cornell.edu

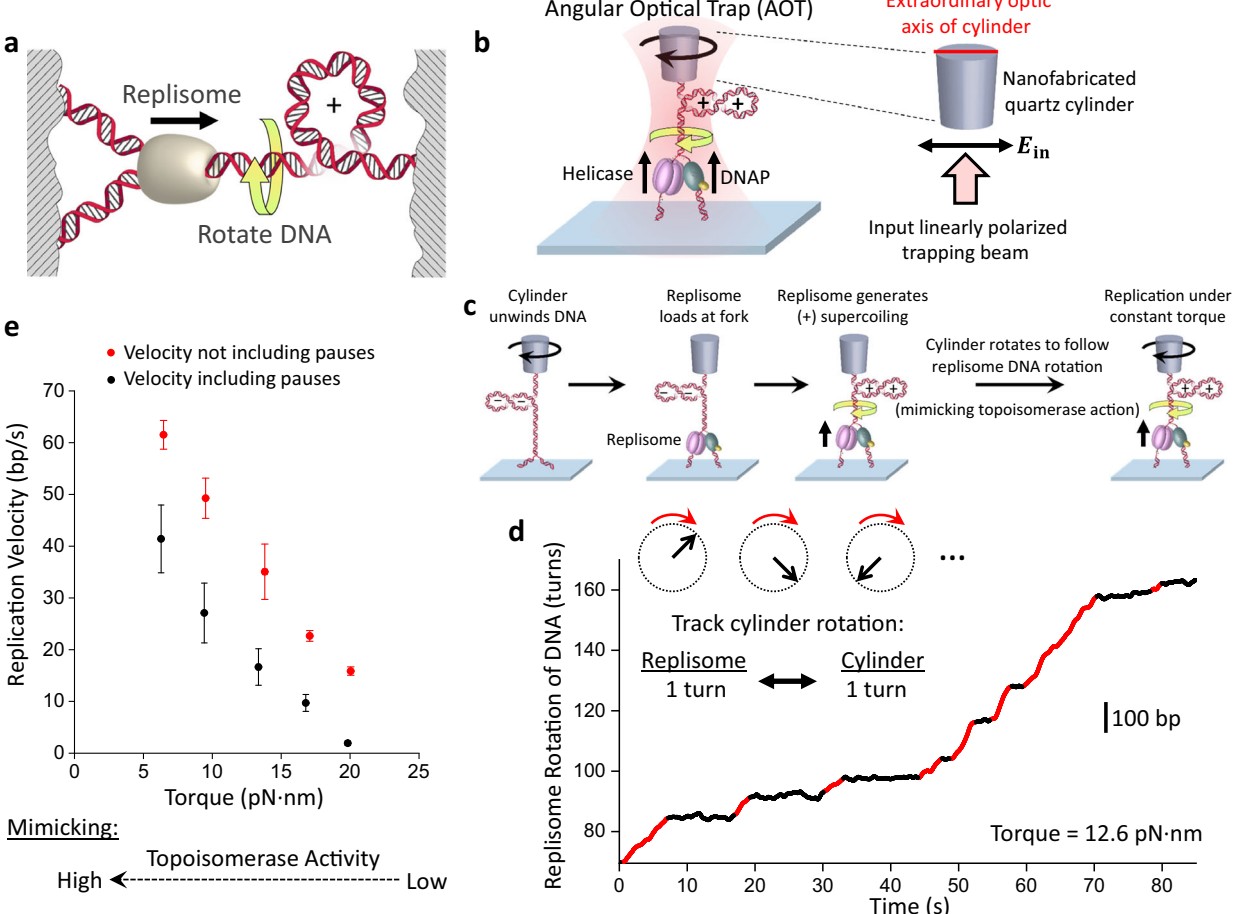

**Fig. 1 | Visualizing replisome rotation of DNA under defined torsion. a** A cartoon depicting replisome rotation of DNA during replication. Replication generates extra turns in the DNA substrates. If these extra turns cannot be fully dissipated at the DNA ends or relaxed by topoisomerases, they will produce (−) torsion in DNA. **b** An experimental configuration to track replisome rotation of DNA using the angular optical trap (AOT) (Methods). A Y-shaped DNA substrate is torsionally constrained between the bottom of an optically trapped quartz cylinder and the surface of a microscope coverslip. Replisome progression generates (+) torsion that rotates the cylinder. The replisome rotation is indicated by the angular orientation of the cylinder, as measured by the cylinder's extraordinary optic axis. This axis tends to align with the polarization of the input trapping beam (right inset), with its orientation precisely detected by the AOT's torque detector. **c** Experimental scheme used to track replisome rotation of DNA. Using the AOT, we

unwind the parental DNA. This introduces (−) supercoils into the DNA substrate, facilitating the opening of the fork and thus the loading of the replisome. After a T7 replisome loads at the fork and replication begins, the replisome rotates the parental DNA, adding (+) supercoils to the parental DNA, causing the DNA to buckle to form a plectoneme. Once a desired torque is reached, the cylinder is rotated to follow the replisome rotation, using a feedback routine to maintain a constant torque on the DNA. **d** An example trace of replisome rotation of DNA under a 12.6 pN·nm torque as visualized via the cylinder (the corresponding Movie is shown in Supplementary Movie 1). Continuous replication (red regions) is interrupted by pauses (black regions). The scale bar provides the conversion to the translocation distance of the replisome. **e** Torque-velocity relation of replication. Each data point is obtained from $N = 14, 14, 15, 16, 15$ individual traces from left to right. The error bars are SEMs. Source data are provided as a Source Data file.

direct methodologies for investigation. Here, we developed an assay to track DNA rotation by T7 replisome and its slowing under torsional stress in real-time.

## Results

### Visualizing replisome rotation of DNA

The fundamental cause for torsional stress during replication stems from the obligatory replisome rotation of the DNA. Understanding the consequences of this rotation requires an experimental method to track the rotational dynamics, which has not been possible for DNA replication. Thus, we developed a method to directly visualize replisome rotation of DNA in real time under a defined torsion. This method was implemented using an angular optical trap (AOT) (Fig. 1b), which is ideally suited to study DNA rotational motion and torsional stress[35–38,41–45]. A defining feature of the AOT is its trapping particle. Unlike conventional optical tweezers which typically trap a polystyrene or silica microsphere, an AOT traps a nanofabricated birefringent quartz cylinder[41] (Supplementary Fig. 1). The AOT accurately

detects the rotational orientation of the cylinder by the polarization state of the transmitted trapping laser beam[41,46] and thus informs the rotational orientation of the attached DNA molecule (Fig. 1b). Here, we have used the cylinder to track the replisome rotation of DNA under a defined torsion (torque).

We enabled this method using the T7 replisome, a model system for studying replication. This minimal system consists of T7 helicase, T7 DNA polymerase (DNAP), and thioredoxin[47,48] (Fig. 1b; Methods). Thioredoxin is a DNAP processivity factor that binds to DNAP to increase its processivity. To begin the experiment, after torsionally anchoring a Y-shaped DNA substrate (Supplementary Fig. 2; Supplementary Fig. 3) between the coverslip and a quartz cylinder held in the AOT, we use the AOT to unwind the DNA which weakens the base pairing interactions of the parental DNA at the fork to facilitate replisome loading (Fig. 1c). Subsequently, with the cylinder angle held constant, the replisome rotates DNA, converting (−) supercoiling to (+) supercoiling which then hinders replication, generating increasing (+) torsion as the replisome proceeds. Once a torque value of interest is

reached, we allow the cylinder to rotate by following the replisome rotation of DNA. In this mode of operation, the torque at the parental DNA is maintained at a constant level by compensatory cylinder rotation, thereby limiting further torsion buildup. Thus, for each turn the replisome rotates the DNA (-10.5 bp), the cylinder also rotates a turn. Because the cylinder's extraordinary optic axis angle can be tracked at exceedingly high spatial and temporal resolution, this method provides unprecedented resolution of replication fork dynamics.

Using this AOT method, we directly visualized fork motion in real-time under a specified torque. In the example curve shown in Fig. 1d (Supplementary Movie 1), we placed the replisome under 12.6 pN·nm of torque to intentionally slow down the replisome for real-time visualization. This rotational trajectory provides detailed information on the replisome motion over a thousand base pairs; the replisome rotates DNA at about 4 turns/s, and the continuous rotation is interrupted by pauses, which reflect a temporary halt in the replication motion, likely due to events such as transient helicase or DNAP dissociation, or DNAP stuttering.

In vivo, topoisomerases relax DNA torsional stress. Thus, DNA torsion depends on the extent of topoisomerase activity. Here, we mimicked the extent of topoisomerase activity by specifying a torque value (Fig. 1e). At a low torque (mimicking high topoisomerase activity), the replisome replicates at a high velocity. As the torque increases (mimicking a decrease in topoisomerase activity), the replisome's velocity after removing the pauses (Methods) decreases with a concurrent increase in pausing, leading to a significantly lower velocity when pauses are included. The resulting torque-velocity relation specifies how the replisome slows down in response to torsion, a relation characteristic of the chemo-mechanical properties of the rotary motor.

Thus, by directly visualizing replisome rotation of DNA, we demonstrate that the replisome is a rotary motor, capable of rotation against torsional stress. Although this rotational motion is inherent to the helical nature of the DNA, our direct visualization highlights that rotation is an inevitable consequence of replication, paving the way for understanding the impact of torsional stress on fork stability.

## The replisome is a powerful rotary motor

In vivo, torsional stress is generated during DNA replication elongation and termination[18-27]. In addition, ( + ) torsion is thought to promote replisome stalling during a head-on conflict between a replisome and an RNA polymerase (RNAP) because both motors generate (+) supercoiling in front, leading to (+) torsion accumulation between them[28-30]. Studies also suggest that there may not be sufficient copies of topoisomerases in the cell to keep up with the torsional stress generated by replication[32,33], resulting in a transient shortage of topoisomerases in regions of the genome. Topoisomerase inhibition leads to replisome fork slowdown, stalling, and fork reversal in dividing cells[18,27,49-51], further demonstrating the crucial need for topoisomerase relaxation of replication-generated torsional stress.

To investigate replisome stalling under torsion, we modified the method depicted in Fig. 1c. Instead of following the replisome rotation of DNA, we restrict the cylinder rotation, thus allowing the DNA substrate to accumulate (+) torsion until the replisome comes to a stall (Fig. 2a). We directly measured the DNA extension, the force, and the torque on the DNA during the process of stalling (Fig. 2b, c) and obtained the fork position from the measured extension (Methods; Supplementary Fig. 4). Figure 2b, c show example traces where the replication fork position is monitored as torsion accumulates. We observed that once both motors are loaded at the fork, the fork moves forward rapidly until it stalls under torsion. We found that the stall torque measured directly using our torque detector agrees well with that of the known DNA buckling torque (Supplementary Fig. 5), indicating that the replisome was sufficiently powerful to buckle the parental DNA into a plectoneme at a stall. Control experiments show

minimal detectable fork movement without replication proteins or dNTPs (Supplementary Fig. 6), and the force on DNA is sufficiently small not to have any detectable impact on the replication rate (Supplementary Fig. 7).

A replisome is a rotary motor that converts chemical energy into mechanical work to rotate DNA. The stall torque is an inherent property of a rotary motor and indicates its ability to move forward against torsional stress, dissociate bound proteins, and reconfigure DNA structures and topology[52,53]. We found the WT replisome stalls at a torque of 21.9 ± 4.4 pN·nm (mean ± SD), making it a powerful rotary motor (Fig. 2b, d). To put this value in perspective, *E. coli* RNA polymerase (RNAP) on its own can generate -11 pN·nm of torque[35], which is sufficient to melt DNA[35,44,45], and a torque of +19 pN·nm has been shown to significantly facilitate the dissociation of histone H2A/H2B from a nucleosome[34]. Therefore, the T7 replisome is about twice as powerful in terms of torque generation capacity and is the most powerful DNA rotary motor studied to date.

It is possible that the T7 replisome torsional capacity simply results from the additive torsional capacities of the two motors at the fork. To investigate this possibility, we measured the stall torque of DNAP (complexed with thioredoxin) without helicase and found it to be −1.5 ± 2.4 pN·nm (Fig. 2d), revealing that DNAP alone has minimal capacity to generate (+) torsion despite bearing some structural and functional similarities to RNAP. To determine the stall torque of the helicase alone, we used a modified Y-shaped DNA substrate containing a ssDNA region for helicase loading (Methods). We found the stall torque to be 1.2 ± 5.0 pN·nm (Fig. 2d), showing that helicase alone also has minimal capacity to generate (+) torsion.

We wondered if the replisome's high torsional capacity is a result of the specific interaction between the C-terminal domain (CTD) of the helicase and DNAP, as this interaction is essential for T7 phage replication[54]. We thus carried out a similar stalling experiment with a replisome containing a $\Delta C_t$ helicase, using a mutant T7 helicase ($\Delta C_t$) that lacks the 17 carboxyl-terminal amino acid residues required for interaction with T7 DNAP[54,55], and we refer to this replisome as the $\Delta C_t$ replisome (Fig. 2c). The $\Delta C_t$ helicase is known to have DNA unwinding activity comparable to the wild-type helicase but does not interact with DNAP and cannot support processive DNAP leading strand replication[54-56]. We found that the $\Delta C_t$ replisome can also work against significant torsion, albeit being less stable upon stalling, with the fork position sometimes undergoing significant reverse movement (Fig. 2c), but it remains a powerful rotary motor, generating a stall torque of 19.4 ± 3.0 pN·nm (Fig. 2d), only marginally smaller than that of the wild-type (WT) replisome. Therefore, these results indicate that the replisome's torsional capacity is primarily due to two motors at the fork.

## Torsion leads to fork regression

To characterize fork stability during a stall, we aligned the fork position of each trace at its maximum value and plotted how the average fork position regresses over time, leading to a concurrent torsion reduction (Fig. 3a). We found the fork of the WT replisome is comparatively stable, with the fork regressing about 80 bp over a minute. In contrast, the $\Delta C_t$ replisome undergoes a more dramatic fork regression of 240 bp in a minute, reflecting a reduced ability to sustain torsional stress and maintain fork stability.

The observed fork regression could result from two distinct mechanisms. Since DNAP has an exonuclease activity, the replisome may reverse translocate by DNAP removing the replicated nucleotides (Fig. 3b, left panel). Alternatively, the torsional stress generated by the replisome could reverse the fork by squeezing the replicated strand off the template DNA, forming a "chicken-foot-like" structure (Fig. 3b, right panel), which bears some resemblance to the chicken-foot structure previously reported under torsional stress[19,20,57]. Such a structure maintains the same number of base pairings in the DNA

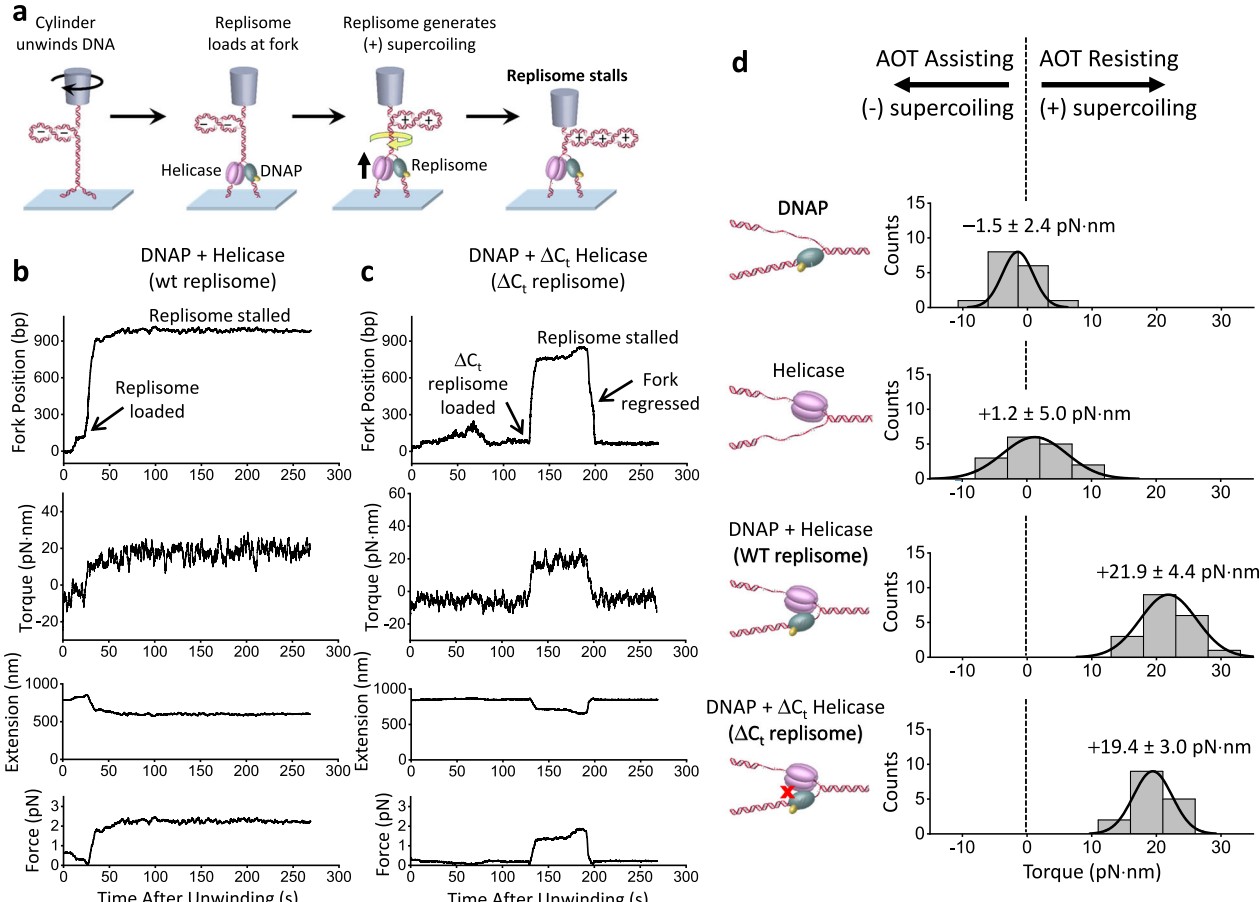

**Fig. 2 | Stalling the replisome under torsion. a** Experimental scheme used to stall a replisome under torsion. This experiment starts in the same way as shown in Fig. 1c, but in this case, the cylinder is not allowed to rotate after the initial unwinding. While the trap height is held constant, the replication fork is initially underwound to facilitate fork opening and loading of the replisome. Upon replisome loading at the fork, the replisome converts (−) supercoiling to (+) supercoils. This leads to the buckling of the parental DNA, resulting in a decrease in DNA extension and an increase in the measured force and torque. As the (+) torsion generated by replication accumulates, the replisome eventually stalls. **b, c** Example traces of WT replisome and a replisome containing a ΔC$_t$ helicase. For each trace, force on the DNA, DNA extension, and torque on the DNA are directly measured.

The fork position is subsequently obtained from the measured extension (Methods; Supplementary Fig. 4). **d** Stall torque of the replisome. Shown are histograms of the measured stall torque for DNAP alone, helicase alone, WT replisome, and the ΔC$_t$ replisome from the number of traces $N$ = 16, 16, 19, and 16, respectively. Each histogram is fit by a Gaussian function, and the mean and SD of the Gaussian are indicated. A negative stall torque indicates that an enzyme can advance if assisted by an unwinding torque that promotes strand separation of the parental DNA, whereas a positive stall torque indicates that an enzyme can move forward even when opposed by an overwinding torque that resists strand separation of the parental DNA. Source data are provided as a Source Data file.

structure during fork regression, making it energetically mobile over a longer distance while releasing torsional stress.

To differentiate between these two possibilities, we stalled the replisome containing a ΔC$_t$ helicase and an exo- DNAP (referred to as ΔC$_t$ exo- replisome; Methods), which lacks exonuclease activity and can only form a chicken-foot-like structure under fork regression. We found that the ΔC$_t$ exo- replisome also displays long-distance fork regression (reversing about 340 bp in a minute), similar to but at a slightly greater extent than the ΔC$_t$ replisome, suggesting that the fork regression observed for the ΔC$_t$ replisome may primarily be a result of the chicken-foot-like structure formation. These findings demonstrate the crucial role of the specific interaction between the helicase and DNAP in stabilizing the replication fork under torsional stress by limiting fork reversal. In the absence of this interaction, the fork may collapse through the formation of chicken-foot-like structures, rendering the replisome inactive.

These results reveal that the fork of a basal replication machinery composed of helicase, DNAP, and thioredoxin is dynamic upon stalling under torsion. Even the WT replisome undergoes moderate fork regression. Since fork regression releases torsional stress, such a regression could be a protective mechanism to tolerate the torsional

stress and prevent genome instability[58–61]. In vivo, fork reversal under stress is further modulated by other enzymes, such as annealing DNA helicases, which regulate fork reversal to safeguard genome integrity[58,62–64].

## Prolonged torsion limits fork restart

In vivo, if the replisome is stalled by torsion, torsion may be relaxed by the arrival of topoisomerases[65,66]. Subsequent replication restart is crucial for maintaining genomic integrity and preventing mutations and is regulated by a multitude of enzymes and factors[60,67]. Here, we investigated what governs replication restart in the presence of the basal replication machinery proteins using the AOT. For these experiments, after the replisome is stalled under torsion for a specified time, we mimic topoisomerase torsional relaxation by unwinding DNA to reduce the torsion and examine the replisome's innate ability to restart replication (Fig. 4a−c). In the torsion relaxation step, we use the AOT cylinder to attempt to unwind DNA to reduce the torque to +6 pN·nm, which is well below the stall torque and requires unwinding of no more than 50 turns if the replisome is inactive during the relaxation. However, in cases where the replisome becomes active and generates a torque greater than

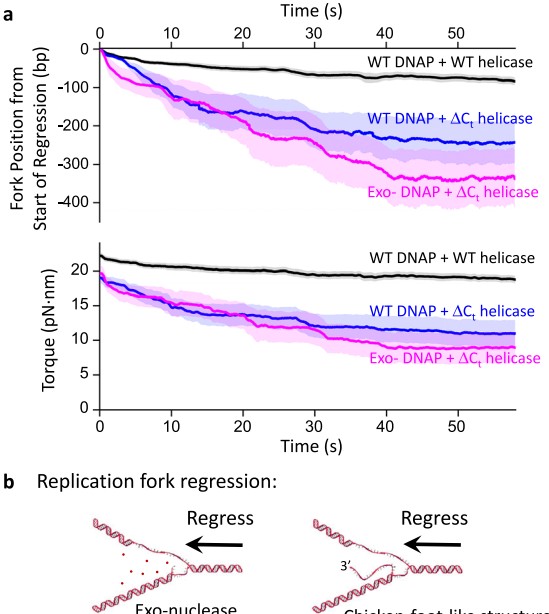

**Fig. 3 | Fork regression dynamics during stalling under torsion. a** Fork regression versus time. All traces are aligned in position and time at their maximum forward fork position. For each type of replisome, we show the mean fork position versus time, with their SEM bracketing the shaded regions. The total number of traces are: 48 (wt replisome), 22 ($\Delta C_t$ replisome), and 12 ($\Delta C_t$ exo- replisome). **b** Cartoons illustrating two possible mechanisms of fork regression. Enzymes bound near the fork are omitted for clarity of the fork configuration. Source data are provided as a Source Data file.

+6 pN·nm, we unwind 50 turns. After torsion relaxation, if the replisome is fully active, it should restart replication, generate torsion, and stall again under torsion. We use the restart stall torque after the unwinding step as a measure of restart efficiency; a fully active replisome after the restart should reach a torque close to its initial stall torque.

We found that when the replisome is initially stalled for 30 s or 60 s, the fork exhibits minimal regression during the stall, and replication restarts efficiently upon torsion reduction before stalling again at a similar stall torque (Fig. 4b, d). However, as the initial stall time increases to 90 s and 120 s, the replication undergoes more significant fork regression during the initial stall, with the restart stall torque reduced from the initial stall torque (Fig. 4b, d). The results demonstrate that prolonged stalling inactivates replication, highlighting the importance of timely torsional relaxation. A delay in topoisomerase arrival in vivo could lead to replication stalling under torsion, rendering the fork less active.

We then examined the role of DNAP-helicase interaction in the replication restart (Fig. 4c, e). We found the $\Delta C_t$ replisome reaches an initial stall torque similar to that of the WT replisome but undergoes more long-distance fork regression during the stall and has a reduced restart stall torque compared to the WT replisome. Thus, the specific interaction between DNAP and helicase not only stabilizes the fork during stalling but also promotes replication restart after stalling.

### Excess DNAP promotes fork restart

We were particularly intrigued by the low restart rate under the 120 s stalling time (Fig. 4b, d), as the inability to restart replication is detrimental to genome stability. To investigate this problem, we must explore a range of experimental conditions. The scope of these experiments motivated us to develop a new method based on magnetic tweezers (MT), which enables parallel measurements of multiple

replisomes under torsion. The experimental procedure for replication restart using the MT (Fig. 5a) is similar to that using the AOT (Fig. 4a). Although this MT method does not allow direct torque measurements, it can readily monitor replication elongation, stalling, and restart, while the torque can still be obtained indirectly from the force since stalling occurs when the DNA is buckled to form a plectoneme according to our AOT data (Supplementary Fig. 5). After allowing the replisome to stall for a specified time, we mimic topoisomerase torsional relaxation by unwinding DNA 100 turns, which can fully relax torsional stress and promote the reversal of the fork regression (Methods). Subsequent replisome restart generates (+) supercoiling and buckles the DNA to form a plectoneme. Under the 1.0 pN force used in these assays, (+) DNA buckling occurs at 12.6 pN·nm torque, which can only be generated by an active replisome (Fig. 2d). Thus, we classify a trace as having restarted if the replisome can (+) buckle the DNA after the unwinding step (Fig. 5b).

Using the MT method, we focused on factors that can facilitate replication restart after a long stall (240 ± 17 s; mean ± SD). We first examined the role of excess DNAP in replication restart (Fig. 5c). If the low restart rate results from DNAP dissociation under stress, then DNAP must reload before replication restarts, and inefficient DNAP reloading could limit the restart. However, the DNAP loading time on the initial DNA substrate is short at 1 nM DNAP (Supplementary Fig. 8), suggesting that reloading of DNAP to a normal fork should not be a limiting factor. Surprisingly, we found a marked increase in the restart fraction from 40% to 85% when the DNAP concentration increased from 1 nM to 100 nM. Significantly, this enhanced restart requires excess DNAP to be present during the initial stall, as increasing DNAP concentration to 100 nM after the prolonged stall is much less effective in promoting replication restart (Fig. 5c, green curve). Thus, the findings demonstrate that prolonged replisome stalling by torsion inactivates replication, which can be effectively circumvented by having excess DNAP.

To further investigate the role of DNAP concentration, we examined the role of the helicase CTD, which has previously been shown to mediate DNAP recruitment[68,69] (Fig. 5c, purple curve). Since the T7 helicase is a hexamer, multiple DNAP enzymes can be recruited to the helicase by its CTD, greatly increasing the effective local concentration of DNAP at the fork (on the order of mM). Indeed, the $\Delta C_t$ replisome, lacking the ability to recruit DNAP, shows minimal increase in the replication restart fraction when the DNAP concentration is increased from 1 nM to 100 nM, indicating a crucial role of the helicase CTD in recruiting DNAP to promote fork restart. This finding lends strong support to the mechanism that excess DNAP helps maintain an active fork. To determine if an excess of other proteins, such as helicase and *E. coli* SSB, involved in replication can play a similar role as excess DNAP in replication restart, we increased the helicase concentration by 10 times from 180 nM to 1.8 μM and found the restart fraction to be minimally affected (Fig. 5d). SSB has been implicated in facilitating fork progression and might play a role in the restart efficiency[70]. We found that adding 200 nM SSB in the reaction does not promote replication restart (Fig. 5d). Thus, our results support a highly significant role of excess DNAP in maintaining an active fork during the stall and facilitating subsequent replication restart after torsional relaxation.

### Gyrase facilitates replication and its restart

The results thus far suggest that topoisomerases are crucial for relaxing DNA torsional stress and maintaining an active replication fork. Although we have mimicked topoisomerase action in the above experiments, a complete understanding of replication under torsion requires a method to track the replication fork position in real time while topoisomerases relax torsional stress. Detecting the replisome movement is challenging because the measured extension results from a combination of replisome movement and topoisomerase relaxation. To enable the detection of replisome movement, we

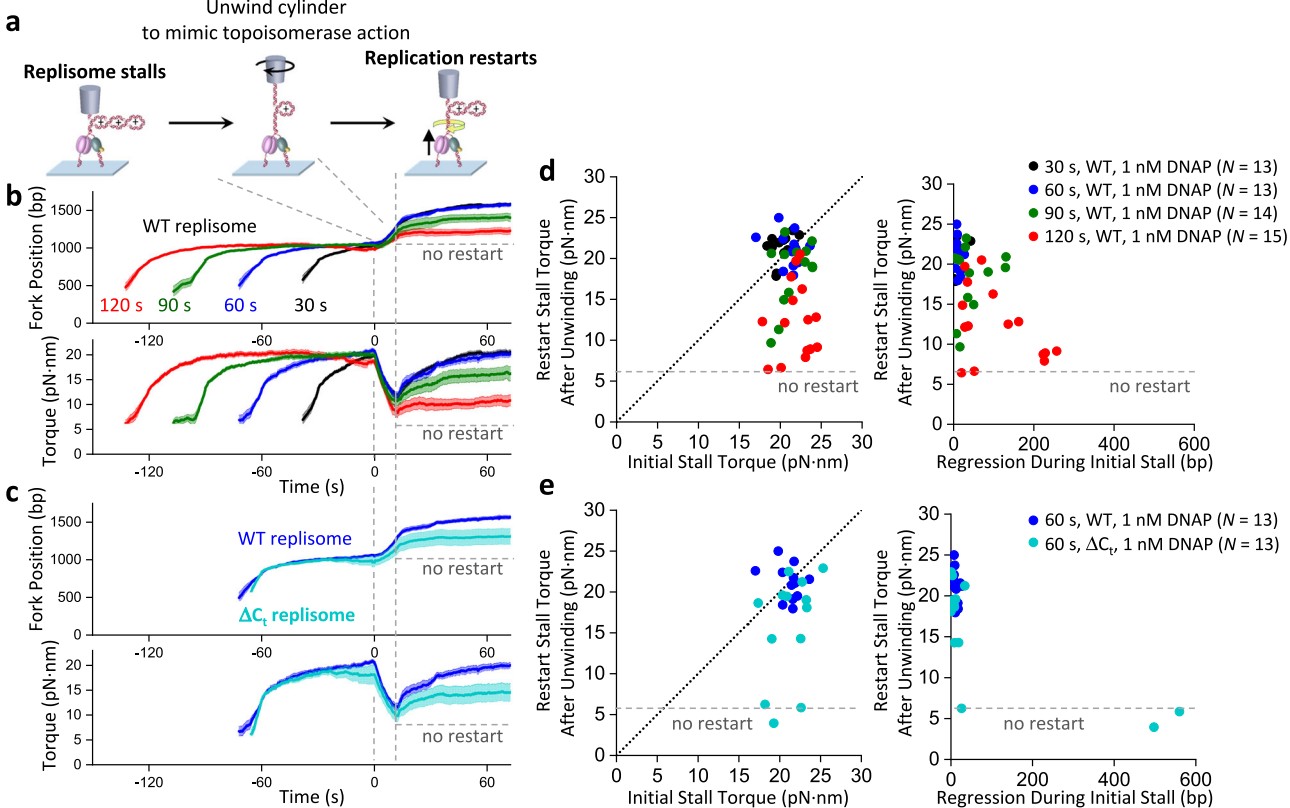

**Fig. 4 | Replication restart after stalling. a** Experimental scheme for restarting replication using the angular optical trap. The replisome is first loaded at the fork and allowed to accumulate (+) supercoiling and stall (as shown in Fig. 2). Subsequently, we unwind the cylinder to release some of the torsional stress, mimicking the action of topoisomerases. After unwinding, replication may resume, leading to the accumulation of (+) supercoiling and stalling of the replisome. **b, c** Replication restart under different stalling times and protein compositions. Shown are the mean curves after aligning all traces at the start of the AOT unwinding. For each condition, the mean curve of all traces is bracketed by the SEM (shaded region). The number of traces is indicated in the relevant figure panel. **d, e** Replication restart characterization from traces used in (**b, c**) respectively. Each data point came from a single trace. The left panel plots show the effectiveness of the restart, characterized by the restart stall torque after the unwinding step. The diagonal dashed lines represent when the restart stall torque is the same as the initial stall torque, indicating an effective restart. The right panel plots show how the restart stall torque correlates with the regression distance during the initial stalling. The restart stall torque showed a negative correlation with the regression distance during initial stall. The Pearson correlation coefficient $r = -0.589$ (modest negative correlation) with $p = 2.84 \times 10^{-6}$ for the combined data points in panel d; $r = -0.753$ (strong negative correlation) with $p = 0.003$ for the $\Delta C_t$ replisome data points in panel e. The statistical tests were two-sided. Source data are provided as a Source Data file.

developed an MT-based method by periodically sampling the fork position (Supplementary Fig. 9). Using this method, we examined how gyrase, an essential topoisomerase in *E. coli*, regulates fork progression (Fig. 6a). In the absence of gyrase, the replication fork stalls after converting the pre-introduced (−) supercoils to (+) supercoils, similar to the behavior shown in Fig. 2b and Fig. 4b. With an increase in the gyrase concentration, the replication speed increases and reaches a greater distance with less stalling. Thus, the presence of sufficient gyrase is essential to ensure steady processive replication.

If there is a transient shortage of gyrase, the replisome may be stalled, and the subsequent arrival of gyrase may restart replication. To investigate the replisome's ability to restart after a delay in gyrase arrival, we conducted experiments by allowing the replisome to stall for a defined duration before introducing gyrase and examining the restart fraction (Fig. 6b). We found that the replisome can readily restart after a short stall duration. As the stall duration increases, the replisome becomes increasingly less efficient in restarting the replication. These observations directly demonstrate the importance of the timely arrival of gyrase to release the torsional stress.

## Discussion
To understand how a replisome responds to torsional stress, we have investigated the T7 replisome containing the basal replication components. Using this system, our findings provide unprecedented

information on replication under torsion, demonstrating that torsion is a strong regulator of replication.

We show that the T7 replisome, composed of DNAP, thioredoxin, and helicase, is a powerful rotary motor, although DNAP or helicase individually is a weak rotary motor (Fig. 2d), likely because helicase alone can slip under the influence of the fork[71] and DNAP alone can also reverse translocate[72]. However, working in conjunction, the two motors at the fork convert the replisome into a remarkably powerful rotary motor. This synergistic cooperation is primarily a result of one motor keeping the fork open for the other motor by limiting reverse fork motion[73-75] and augmenting DNA breathing at the fork[76,77]. Unlike RNAP, which holds both DNA strands within the motor, the replisome splits apart the two strands, with each motor tracking one strand[48]. This configuration can create a larger lever arm for torque generation around the parental DNA's central axis, akin to using a corkscrew with a more extended handle to screw into the cork of a wine bottle.

The torsional capacity of the replisome may be particularly relevant in regions that are difficult to replicate, such as those involving a head-on transcription-replication conflict. Although the replisome has been found to win in this conflict ultimately[78,79], the mechanism remains unclear. The differential torsional capacities of the two machineries may play a role in how a replisome wins the conflict. Our studies show that the T7 replisome can generate 22 pN·nm of torque (Fig. 2d), twice that of *E. coli* RNAP[35]. Thus, in vivo, the (+) torsion

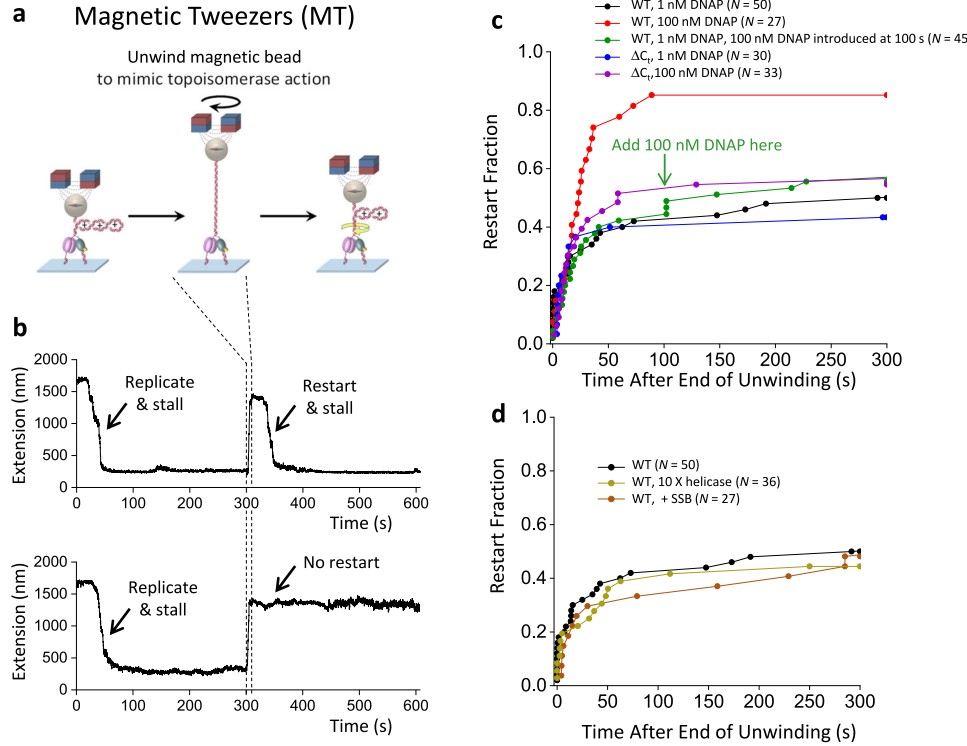

**Fig. 5 | Replication restart after a long stall. a** Experimental scheme for replication restart using magnetic tweezers. As with the AOT experiments in Fig. 4, the replisome is first allowed to load and accumulate (+) supercoiling and stall. Subsequently, we unwind the magnetic bead to release all torsional stress, mimicking effective torsional relaxation by topoisomerases. After unwinding, replication may resume, leading to the accumulation of (+) supercoiling and stalling of the replisome. **b** Two example traces. DNA tethers are held under 1.0 pN. In each trace, the initial replication leads to the generation of (+) torsion, as evidenced by an extension decrease. Continued replication should eventually stall the replisome. The subsequent unwinding of the DNA fully relaxes the (+) torsion, increasing the DNA extension. Following this unwinding step, the ability of the replisome to restart is examined. The top trace shows an example of replisome being able to restart, as revealed by the extension decrease. The bottom trace shows an example of the replisome being unable to restart, where the extension remains essentially unchanged. **c** The effect of DNAP concentration on replication restart fraction. The green arrow indicates the time when we introduced 100 nM DNAP together with helicase into the reaction for the green curve. **d** The effect of helicase concentration and the presence of SSB on replication restart fraction. The olive curve was obtained by increasing the helicase concentration to 10 X of that used in the standard condition (Methods). The maroon curve was obtained by including 200 nM SSB in the standard condition. Source data are provided as a Source Data file.

between the two machineries could stall the RNAP first while permitting the replisome to continue moving forward. This could occur well before any physical encounter of the replisome with RNAP, as torsion in the DNA acts over distance. Even so, a delay in topoisomerase arrival at the fork could accumulate (+) torsion and slow down and eventually stall the replisome, decreasing the ability of the replisome to restart (Fig. 6b). These findings highlight the crucial role of topoisomerases in timely torsion relaxation.

We show that an active fork requires interactions between the helicase and DNAP. We previously found that the T7 helicase on its own can readily slip backwards[71] but becomes highly processive when complexed with a non-replicating DNAP, and this interaction, mediated via the CTD of the helicase, is sufficiently strong to allow helicase to drag the non-replicating DNAP along the DNA[56]. The current work shows that this interaction also stabilizes the fork, prevents long-distance fork reversal during stalling (Fig. 3a), and facilitates subsequent fork restart (Fig. 4c, e). While this interaction ensures fork stability, it may also present a potential problem for replication restart if the DNAP in the replisome becomes inactive during stalling.

We made a surprising discovery that the T7 replisome resolves this potential issue when excess DNAP is present during stalling (Fig. 5c). However, the exact mechanism remains unknown. The presence of the excess DNAP may allow the DNAP associated with the replisome to stay bound by limiting its dissociation. Since excess helicase or the presence of SSB does not increase restart efficiency, it is also possible that the restart may be mediated by DNAP exchange to

replace an inactivated DNAP. Previous work found DNAP exchange could occur under no replication stress and suggested DNAP exchange as a mechanism of enabling replication to overcome replication stress for subsequent replication restart[68,69,80–87].

Although our work focuses on the T7 replisome, the experimental approaches established here can be broadly applied to other replication systems. While the helicases in the T7, the T4, and the bacterial replisomes are located on the lagging strand, CMGs in eukaryotic replisomes are positioned on the leading strand ahead of the DNAP[88]. Despite these differences, these replisomes all possess specific interactions between the helicase and the replicative DNAP. These replisomes may differentially regulate their torsional generation capacity, fork regression under torsion, and the subsequent fork restart upon torsional relaxation. This work creates many new opportunities to investigate replication dynamics under torsion.

## Methods

### Protein purification

The M64L mutant of T7 helicase gp4 (T7 gp4A') and its C-tail deletion mutant with 17 C-terminal amino acid residues deleted (gp4A' $\Delta C_t$) were purified from *E. coli*[72,89]. Introduction of M64L mutation inactivates a second start codon in T7 gp4 ORF and prevents expression of gp4B. *E. coli* cells expressing these proteins were lysed by freeze-thaw cycles in lysis buffer (50 mM Tris-HCl, pH 7.5, 100 mM NaCl, 1 mM EDTA, 1 mM DTT, 10% sucrose, 1 mM PMSF and 0.4 mg/ml lysozyme). Polyethylenimine (PEI) was added to the lysis supernatant followed by

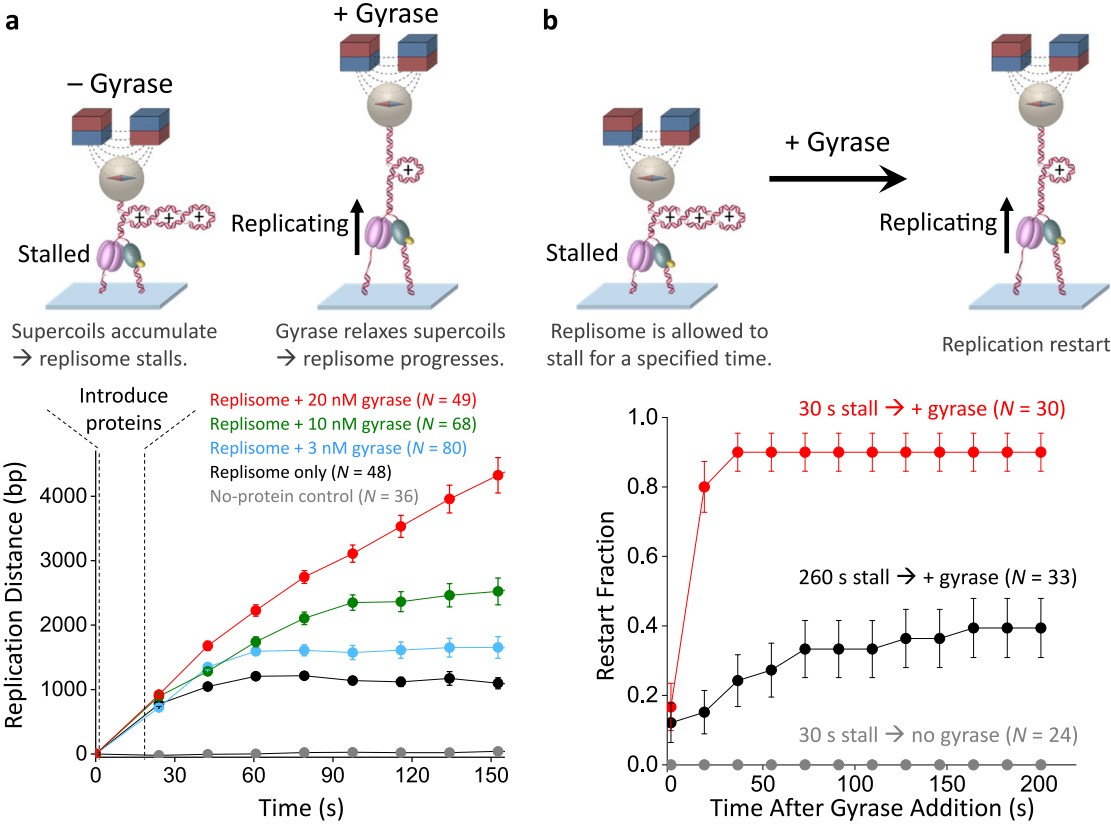

**Fig. 6 | Gyrase facilitates replication progression and restart under torsion.**
**a** Gyrase facilitates replication progression under torsion. The top cartoons illustrate replication progression with or without *E. coli* gyrase under 1.5 pN force using magnetic tweezers. In the absence of gyrase, replication accumulates (+) supercoiling and stalls. In the presence of gyrase, torsional stress is relaxed, allowing continuous replication. The bottom plots show the replisome progression over time under different gyrase concentrations, using a method that determines the fork position based on the torsional-mechanical properties of the DNA template

(Methods; Supplementary Fig. 9). The error bars are SEMs. **b** Gyrase facilitates replication restart after the replisome stalls under torsion. The top cartoons show experimental procedures. After the replisome stalls under torsion for a specified time duration, 20 nM gyrase is introduced to relax the torsion stress. A replisome is considered to have restarted if it has progressed for at least 2000 bp. The bottom plots show the ability of a replisome to restart after gyrase introduction. The error bars are SEMs. Source data are provided as a Source Data file.

ammonium sulfate precipitation. The precipitates were solubilized and further purified on phosphocellulose (P-11 resin) and DEAE Sephacel chromatography columns.

Exonuclease deficient mutant of T7 gp5 DNA polymerase (D5A, E7A) was also purified[90]. Briefly, *E. coli* cells expressing T7 gp5 Exo- were lysed in buffer consisting of 50 mM Tris-HCl, pH 8.1, 2.5 mM EDTA, 1 mM β-mercaptoethanol, 0.1 mM DTT, 0.3 mg/mL Lysozyme, 150 mM NaCl and 0.1% sodium deoxycholate. NaCl concentration was gradually increased to 0.5 M while gently stirring the lysate on ice. Lysate was cleared by centrifugation at 39000 × *g* and the protein was precipitated using 0.5% final concentration of PEI, pH8. Protein was purified by ammonium sulfate precipitation. Precipitates were dissolved, dialyzed to remove excess salt, and further purified on phosphocellulose (P-11 resin) and DEAE sephacel chromatography columns.

*E. coli* SSB was purified from *E. coli*[91]. Briefly, *E. coli* cells expressing SSB were lysed, and the lysate was cleared by centrifugation at 13000 × *g*. PEI was added to the cleared lysate to precipitate the protein. Precipitants were extracted in buffer containing 0.4 M NaCl. Protein was further purified by ammonium sulfate precipitation followed by Q-sepharose column chromatography.

Wild-type T7 gp5 DNA polymerase was purchased from New England Biolabs (NEB, Ipswich, MA). *E. coli* thioredoxin was purchased from Sigma-Aldrich (St Louis, MO).

*E. coli* gyrase was purified. Full-length GyrA and GyrB were cloned into a pRSF-1b vector [Novagen] with 14X His and SUMO tag at the

N-terminus. The GyrA and GyrB constructs were transformed into Rosetta™(DE3) pLysS cells, grown in 2 L of 2xTY media at 37 °C, and expressed at OD 0.8 for 4 h with 0.5 mM IPTG. Cells were harvested, resuspended in 100 mL of buffer-A [50 mM HEPES pH 8.0, 40 mM imidazole pH 8.0, and 10% glycerol along with protease inhibitors (1 mM PMSF, 1 μg/ml pepstatin A, and 1 μg/ml leupeptin)] containing 1000 mM NaCl (A1000), and flash frozen. To purify full-length GyrA and GyrB, cells were thawed and lysed by adding lysozyme (2 mg/mL), incubating on ice for 30 mins, and then centrifuged at 39190 × *g* for 40 mins. Clarified lysate was next passed through a HisTrap-HP 5 mL column equilibrated in Buffer-A1000. After washing with 10 column volumes (CV) of Buffer-A1000 and an additional 10 CV Buffer-A containing 150 mM NaCl (A150), protein was step eluted using buffer-A150 with 500 mM Imidazole directly onto an A150-equilibrated HiTrap-Q 5 mL column. Hitrap-Q column was washed with 10 CV of buffer A150, and protein was eluted using a gradient between A150 and A1000. Peak fractions containing the protein were collected, and the His-SUMO tag was removed by overnight digestion at 4 °C using in-house purified His-tagged SUMO protease. The mixture was then dialyzed for 16 h against Buffer A1000 to remove excess imidazole and any uncleaved material and the tagged protease were separated from the protein by re-passing the solution over a HisTrap column equilibrated in Buffer A1000, collecting the flowthrough, and concentrating using Amicon™ Ultra-15 Centrifugal Filter Units (30 kDa cutoff). The protein was further purified by loading onto a HiLoad Superdex-200 column equilibrated with a buffer containing 500 mM KCl, 50 mM HEPES pH 8.0,

1 mM EDTA, 1 mM TCEP and 10% glycerol (S500). Peak fractions, assessed using SDS-PAGE, were then collected, concentrated, and adjusted to 20% glycerol prior to freezing in liquid nitrogen.

## DNA substrates

We conducted the majority of experiments using a Y-shaped DNA substrate[92–95] (Supplementary Fig. 2), which is composed of a pre-formed replication fork with a pair of ~2.2 kb symmetric DNA strands upstream of the fork and ~5 kb of parental DNA. Each end of the Y-shaped template is ligated to a ~500 bp multi-labeled tethering adaptor[38,45,96]. The upstream DNA is synthesized via PCR from pLB601 with a shared reverse primer (ATCGTTACTGAGACCCTGATTTAACA AAAATTTAACG), a forward primer (ATTACCAGATCGCTGGAAGCTA-GAGTAAGTAGTTC) for the leading strand, and a reverse primer (TAATGGTTACCGGGAAGCTAGAGTAAGTAGTT) for the lagging strand. The parental DNA is amplified via PCR from pRL574 with a forward primer (ACTGCACCTAGTGATCCGAAGGACAACCTGTTC) and a reverse primer (TGATGCCACCCAGGCTAAGCCCTCCCGTATCG-TAG). The pre-formed replication fork was made of 4 ssDNA oligos (IDT), which are annealed and ligated to upstream and parental DNA[92]. These primers and ssDNA oligos were from Integrated DNA Technologies (Newark, NJ). The pre-formed replication fork DNA sequence does not allow fork regression past the initial fork position. Finally, the upstream DNA strands are nicked to ensure the daughter strands are torsionally relaxed.

For the helicase-only experiments, we used a Y-shaped DNA substrate that is nearly identical to the template described above except containing a small region of ssDNA on the lagging strand near the fork. The only difference is that the pre-formed replication fork was made of 3, rather than 4, ssDNA oligos (IDT). After the 3 ssDNA oligos are annealed, a 27-nt ssDNA region is naturally formed at the fork for helicase loading.

## Experimental conditions

Single molecule tethers were formed in a nitrocellulose coated sample chamber[96–98]. The sample chamber surface was functionalized with 25 ng/µL antidigoxigenin (Vector Labs MB-7000-1, lot # ZH0728, ZJ0203, and ZK0920) and then passivated with 25 mg/mL β-casein (MilliporeSigma C6905). DNA replication substrates were then incubated at a concentration of 5 pM. Quartz cylinders or magnetic beads (Dynabeads MyOne Streptavidin T1, ThermoFisher 65601) were introduced. The chamber was washed with the replication buffer (50 mM Tris-HCl pH 7.5, 40 mM NaCl, 1.5 mM EDTA, 10% glycerol, 8 mM MgCl$_2$, 1 mM dNTPs, 2 mM DTT, 0.5 mg/mL β-casein; for the experiments with gyrase, the same buffer was used except with 9 mM MgCl$_2$ and 1 mM ATP) before flowing in the protein of interest. All experiments were performed in the replication buffer.

Unless stated otherwise, replication was conducted under the following conditions: (1) WT replisome with 1 nM wt T7 gp5 DNA polymerase, 180 nM wt T7 gp4A' helicase (monomer), and 100 nM thioredoxin; (2) ΔC$_t$ replisome with 1 nM wt T7 gp5 DNA polymerase, 45 nM gp4A' ΔC$_t$ helicase (monomer), and 100 nM thioredoxin; (3) ΔC$_t$ exo- replisome with 1 nM exo- gp5 DNA polymerase, 45 nM gp4A' ΔCt helicase (monomer), and 100 nM thioredoxin; (4) wt replisome in the presence of gyrase with 1 nM wt T7 gp5 DNA polymerase, 180 nM wt T7 gp4A' helicase (monomer), 100 nM thioredoxin and *E. coli* gyrase of indicated concentrations (tetramer).

We found that ΔC$_t$ helicase can bind to DNA 4X faster than wt helicase, as determined using our standard helicase unwinding assay[71,99]. We thus decreased ΔC$_t$ helicase concentration by 4X from the wt helicase concentration. In experiments that require 100 nM wt T7 gp5 DNA polymerase, thioredoxin concentration was raised to 1000 nM.

## Single-molecule methods

Our lab previously developed the angular optical trap (AOT) for simultaneous measurements of force, extension, torque, and rotation of DNA[41,46,100,101]. A defining feature of the AOT is its trapping particle, a nanofabricated quartz cylinder (Fig. 1b; Supplementary Fig. 1). The angular orientation of the cylinder is controlled by the trapping laser polarization, while torque on the cylinder is directly measured by the change in the angular momentum of the laser after interaction with the cylinder. At the beginning of each experiment, the trap center of the AOT was set to be ~800 nm above the surface of the cover glass. The tether was unwound by 40 turns which (−) supercoils the DNA and applies (−) torsion to the fork. This (−) torsion weakens the base pairing interactions of the parental DNA at the fork and assists replisome loading. Once the replisome loads at the fork, replication proceeds rapidly, generating (+) supercoiling in the parental DNA. The stall torque for each trace is defined as the maximum torque measured within 60 s after the start of stalling replication.

Magnetic tweezers (MT) experiments were performed on our custom-built instrument[38,96,97]. In this instrument, a pair of magnets apply a constant force to a field of DNA tethers via magnetic beads, and rotation of the magnets rotates the beads to introduce supercoiling to the DNA molecules. During a replication restart experiment (Fig. 5c-d), the force on a Y-shaped DNA substrate is kept at 1.0 pN. After the DNA tethers are unwound by 30 turns, replication start is evidenced by an extension decrease, consistent with (+) supercoil accumulation. At 1.0 pN force, when the DNA is unwound, the extension remains rather flat as the DNA undergoes the melting transition[35,44,45] (Supplementary Fig. 3); whereas when the DNA is overwound, the extension decreases linearly once the DNA is buckled to form a plectoneme[42,45], and the buckling torque is 12.6 pN·nm (Supplementary Fig. 3d), which is greater than the torque that DNAP alone or helicase alone can generate and can only be generated by an active replisome. This break of symmetry provides a convenient way to identify an active replisome by an extension decrease. Continued replication eventually stalls the replisome shortly after the magnetic bead contacts the surface. The extra turns added by replication upon stalling are estimated to be 51 turns based on the stall torque of 22 pN·nm using the torsional stiffness of the buckled DNA and plectonemic DNA[45]. We then wait for a specified amount of time before unwinding DNA at 10 turn/s by 100 turns, which should remove all (+) turns added by replication, resulting in an increase in the DNA extension. Subsequent replication restart is evidenced by an extension decrease.

During a replication with gyrase experiment (Fig. 6; Supplementary Fig. 9), the force on a Y-shaped DNA substrate is kept at 1.5 pN. To start replication, we unwind the tethers 30 turns in the presence of 1 nM DNAP and wait for 60 s before replacing the buffer with the standard replisome condition and the gyrase. Immediately after the introduction of the proteins, replication distance was probed every 18 s. At each detection point, an extension-turn curve was acquired by unwinding DNA tethers at 30 turn/s by 100 turns and then rewinding at 30 turn/s by 100 turns. The 100 turns unwinding is sufficient to bring the tethers to their maximum extensions, which indicates the length of the remaining parental DNA. The unwinding and rewinding speeds are fast to minimize any perturbation to the replication.

## Replication pause analysis

The replication velocity at a constant torque is obtained from the replisome position versus time plots such as shown Fig. 1d. Replisome activity is monitored for 60 s or until the fork regresses at least 100 bp. The velocity including pauses is a linear fit to the rate of replication during this period. To detect pauses, the position versus time plot after being smoothed by 0.5 s Gaussian filter is used to generate the dwell time histogram versus position[102–104]. A pause is detected and removed when the dwell time falls below 0.1 s/bp. The pause-free velocity of

each trace is a duration weighted mean of the linear fits to position versus time of regions of activity between pauses.

### Replication fork position determination using the AOT

The real-time fork position shown in Fig. 2 is determined from the extension of the DNA tether (Supplementary Fig. 4). Before each experiment, the DNA extension is measured as a function of turns applied by the cylinder (Supplementary Fig. 4a, red curve). This calibration curve is then used to convert the DNA extension to the number of turns introduced by the replisome during replication. The replicated turns are then converted to the base pairs replicated by multiplying by the helical pitch of DNA (10.5 bp/turn).

### Reporting summary

Further information on research design is available in the Nature Portfolio Reporting Summary linked to this article.

## Data availability

All data supporting the findings in this paper are provided in the main manuscript and its Supplementary files. Source data are provided with this paper.

## Code availability

Data analysis routines used to process and generate plots are publicly available on GitHub repository (https://github.com/WangLabCornell/replication/)[105].

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

## Acknowledgements

We thank members of the Wang Laboratory for helpful discussion and comments. We especially thank Dr. P.M. Hall for helpful advice on DNA substrate design and Drs. J.E. Peters and M.B. Smolka for helpful comments. This work is supported by the National Institutes of Health grants R01GM136894 (to M.D.W.), R35GM118086 (to S.S.P.), and R37GM071747 (to J.M.B.). M.D.W. is a Howard Hughes Medical Institute investigator. This work has been performed in part at the Cornell NanoScale Facility, a member of the National Nanotechnology Coordinated Infrastructure (NNCI), which is supported by the National Science Foundation (Grant NNCI-2025233).

## Author contributions

X.J., J.T.I., X.G., and M.D.W. designed single molecule assays. X.J. prepared DNA substrates. A.S. and S.S.P. purified and characterized T7 gp4A′, T7 gp4A′ ΔCt, T7 gp5 exo-, and *E. coli* SSB proteins. F.R. and J.M.B. purified and characterized gyrase; X.G. and J.T.I. updated the AOT for replication assays. X.J., X.G., and S.Z. performed single-molecule experiments. Y.H. fabricated quartz cylinders. Y.H., X.G., and X.J. calibrated the cylinders. X.J. and J.T.I. analyzed data. M.D.W. wrote the initial draft. All authors contributed to manuscript revision. M.D.W. supervised the project.

## Competing interests

The authors declare no competing interests.
