## [Transparent Peer Review file · Nature Communications]

Torsion is a Dynamic Regulator of DNA Replication Stalling and Reactivation

Corresponding Author: Professor Michelle Wang

Version 0:

Reviewer comments:

Reviewer #1

(Remarks to the Author)

The authors have done an excellent job responding to our criticisms and we recommend publication.

(Remarks on code availability)

Reviewer #2

(Remarks to the Author)

In their revised manuscript, the authors further discussed how torsional stress and temporary lack of topoisomerase can cause replication stress in physiological conditions. They also reanalyzed their data and found that the inability to restart replication correlates with the extent of fork regression. They removed the claim that DNAP exchange facilitates fork restart, as they did not have compelling evidence, and instead suggested this as a possibility in the discussion. They performed negative controls to show that the activity observed in their single-molecule experiments is indeed due to nucleotide-dependent replisome activity. They also addressed my other minor comments. I am happy to recommend the manuscript for publication in Nature Communications.

(Remarks on code availability)

Reviewer #3

(Remarks to the Author)

The authors have performed significant additional work to address most criticisms of this and the other reviewers, and the work seems now definitively more supportive of the main proposed conclusions.

The addition of gyrase in their system is a significant addition to the work, which validates the relevance of their findings and could prime important follow-up experiments – in their lab and by others - with more complete and less simplified systems.

I remain convinced that the potential of this system will reveal in full once the authors will manage to add crucial cofactors that play key roles in replication fork stalling, remodelling and restart, and that some of their conclusions may significantly change once this will be technically achieved. Some of the proposed conclusions are plausible based on their current setup, but may also be misleading, when attributing to certain intermediates specific physiological or pathological relevance.

The discussion on fork regression (page 7) is surely more balanced in the revised manuscript, although it may be worth commenting that also the addition of enzymatic activities specifically mediating restart – besides those mediating reversal – may drastically change the kinetics observed by the authors, with profound implications on which of those intermediates is in fact linked with fork inactivation.

With these limitations in mind, I anyway consider the revised manuscript as a remarkable experimental tour-de-force and an important step forward integrating topological constraints in an in vitro replication system, worth publication in Nature Communications.

(Remarks on code availability)

Response to Reviewers

Re:

“Torsion is a Dynamic Regulator of DNA Replication Stalling and Reactivation” by Jia et al.

We greatly appreciate the time and effort of all three Reviewers and sincerely thank them for their thoughtful and constructive feedback throughout the review process. We are pleased that all three reviewers found the revised manuscript satisfactory and had no further concerns (see their comments below).

Reviewer #1 (Remarks to the Author):

The authors have done an excellent job responding to our criticisms and we recommend publication.

Reviewer #2 (Remarks to the Author):

In their revised manuscript, the authors further discussed how torsional stress and temporary lack of topoisomerase can cause replication stress in physiological conditions. They also reanalyzed their data and found that the inability to restart replication correlates with the extent of fork regression. They removed the claim that DNAP exchange facilitates fork restart, as they did not have compelling evidence, and instead suggested this as a possibility in the discussion. They performed negative controls to show that the activity observed in their single-molecule experiments is indeed due to nucleotide-dependent replisome activity. They also addressed my other minor comments. I am happy to recommend the manuscript for publication in Nature Communications.

Reviewer #3 (Remarks to the Author):

The authors have performed significant additional work to address most criticisms of this and the other reviewers, and the work seems now definitively more supportive of the main proposed conclusions.

The addition of gyrase in their system is a significant addition to the work, which validates the relevance of their findings and could prime important follow-up experiments – in their lab and by others - with more complete and less simplified systems.

I remain convinced that the potential of this system will reveal in full once the authors will manage to add crucial cofactors that play key roles in replication fork stalling, remodelling and restart, and that some of their conclusions may significantly change once this will be technically achieved. Some of the proposed conclusions are plausible based on their current setup, but may also be misleading, when attributing to certain intermediates specific physiological or pathological relevance.

The discussion on fork regression (page 7) is surely more balanced in the revised manuscript, although it may be worth commenting that also the addition of enzymatic activities specifically mediating restart – besides those mediating reversal – may drastically change the kinetics observed by the authors, with profound implications on which of those intermediates is in fact linked with fork inactivation.

With these limitations in mind, I anyway consider the revised manuscript as a remarkable experimental tour-de-force and an important step forward integrating topological constraints in an in vitro replication

system, worth publication in Nature Communications.